# Design of the Polarization-Independent Wavelength Multiplexing Holographic Metasurface

**Tianyu Zhao** [1,2] ⬤, **Yihui Wu** [1,*], **Yi Xing** [1,2], **Yue Wang** [1], **Jie Wu** [1,2] and **Wenchao Zhou** [1]

1   State Key Laboratory of Applied Optics, Changchun Institute of Optics, Fine Mechanics and Physics, Chinese Academy of Sciences, Changchun 130033, China
2   University of Chinese Academy of Sciences, Beijing 100039, China
*   Correspondence: yihuiwu@ciomp.ac.cn

**Abstract:** Metasurface regulates the polarization, phase, amplitude, frequency, and other characteristics of electromagnetic waves through the subwavelength microstructure. By using its polarization characteristics, it can realize the functions of optical rotation and vector beam generation. It is the most widely used method of regulation. However, parallel optical manipulation, imaging, and communication usually require polarization-insensitive focused (or vortex) arrays of beams, so polarization-independent wavelength multiplexing optical systems need to be considered. In this paper, the genetic algorithm combined with the computer-generated hologram (CGH) is used to control the transmission phase of the structure itself, and on the basis of wavelength multiplexing, the corresponding array of focused or vortex beams without the polarization selection property is realized. The simulation software results show that the method has a huge application prospect in optical communication and optical manipulation.

**Keywords:** reconfigurable metalens; CGH; polarization independent





## 1. Introduction

Because of the advantages of fast, parallel, and simultaneous processing, array beams are widely used in imaging and optical communication. In the field of communication, the vortex beam array can not only improve the security of information but also improve the capacity of information [1–5]. The topological charge carried by the orbital angular momentum and the circular distribution of light intensity can provide torsional torque and gradient force to tiny particles, which can be applied to the control of microscopic particles [6,7]. The vortex beam array can capture more particles at the same time and change the distribution and mode of captured particles through the loading of wavelength reuse and other functions. In the field of imaging, the multiplex-focused beam is more convenient for us to control and compare the beams of different channels, which is of great significance to simplifying the imaging system [8–10].

Currently, beam array generation mostly makes use of beam splitters [11,12], lens arrays [5–16], diffraction optical elements [17,18], etc. The incident beam is split into several beams, and it has a variety of applications in optical communication, imaging, and other domains. However, the manufacturing process of traditional optical components is typically very complex, while the corresponding optical system is also relatively large, making miniaturization of the structure difficult and limiting its application in the field of integrated optics. At the same time, for some complex application fields, the optical components cannot be modulated, which also limits the flexibility of their application. In recent years, research based on metasurface fields has attracted the attention of a large number of researchers.

The metasurface is a periodic or aperiodic array composed of subwavelength structures. The corresponding optical response can be obtained by controlling the geometric

parameters of the unit structure in order to realize control of the incident electromagnetic wave [19–23]. Compared with traditional optical lenses, metasurfaces introduce an abrupt phase, which no longer depends on phase accumulation during propagation and has the advantages of miniaturization and easy manipulation. At present, the metasurface has successively realized the functions of a planar lens [24], a structured beam generator [25], optical holography [26], and polarization control [23,27–29].

For the realization of the array beam function, in 2016, Mehmood et al. used spatial multiplexing to generate a multifocus vortex beam [4]. In 2019, Lv et al. proposed a multifocal metal lens based on an optical metasurface composed of a subwavelength grating etched on a silver film [30]. However, the realization of most metasurface arrays still depends on the geometric phase structure based on polarized light. In order to further improve energy efficiency and simplify the optical system, especially for non-polarized light sources, we need to design a polarization-independent holographic metasurface. For example, fluorescence-labeled single-molecule detection technology, especially the study of multicolor single-molecule fluorescent arrays and biomolecular chirality, has a significant demand for multicolor unpolarized array-focused beams [31–33]. In addition, high-energy density vortex beams are required for optical communication and optical particle manipulation, so polarization-independent holographic vortex array beams will have significant advantages [34–36].

In this paper, a multi-channel array beam generator with wavelength multiplexing is proposed by combining the genetic algorithm and the CGH algorithm. The array beam reconstruction can be realized by adjusting the wavelength of the incident light. Moreover, because we use the propagation phase of the structure itself, the structure has the characteristic of polarization independence. In the same field of view, the wavelength-multiplexed array can avoid the influence of incident beam splitting and multi-wavelength chromatic aberration compared with the splicing of multiple metasurfaces, which has a great application prospect.

## 2. Unit Structure and Principle of the Holographic Metasurface

Wavelength multiplexed holographic metasurface refers to the formation of arrays of spots at different positions in the focal plane when light waves of different wavelengths are incident, as shown in Figure 1 (taking wavelengths R, G, and B as examples).

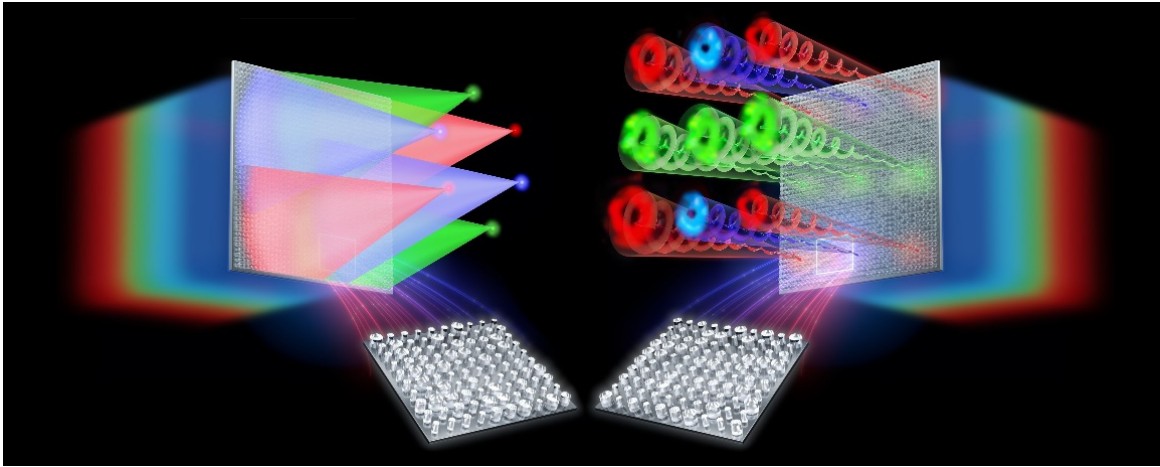

**Figure 1.** Schematic diagram of holographic metasurface for RGB three-wavelength multiplexing.

The phase shift of our designed metasurface, which consists of a properly arranged cylindrical dielectric nanopillar structure with rotational symmetry, depends on the propagation phase with the polarization-independent property. Therefore, to match the calculated ideal phase, we need to build a phase library of the nanopillar. Finite-difference time domain (FDTD) simulations are performed to acquire phase shifts of nanopillars for 2 nm

increment of each geometric parameter by using the commercial software package "FDTD Solutions" (Lumerical Inc., Vancouver, BC, Canada). Periodic boundary conditions are applied along the x and y axes, and the perfectly matched layers (PMLs) are applied in the z direction. The sweeping scope of lengths' diameter of the nanopillars covers 80 nm to 310 nm with fixed height H = 600 nm, while the properly arranged array of nanopillars has a lattice constant of Px = Py = 320 nm. It can be seen from Figure 2a that when the period p = 320 nm, the obtained propagation phase library can cover the phase distribution of $0 \sim 2\pi$ and meet our requirements for phase matching. The specific optimization process is shown in Figure 3.

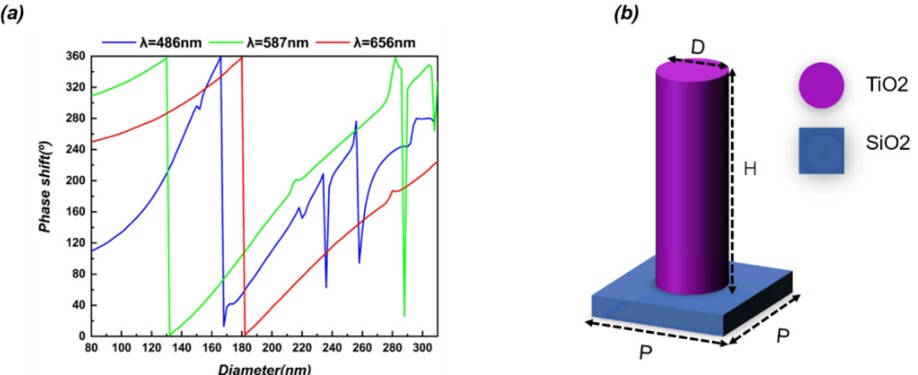

**Figure 2.** (**a**) Phase distribution of the TiO$_2$ nanopillars unit cell; (**b**) Designed nanopillars with a period of 320 nm and a height of 600 nm on SiO$_2$ substrate at RGB wavelengths.

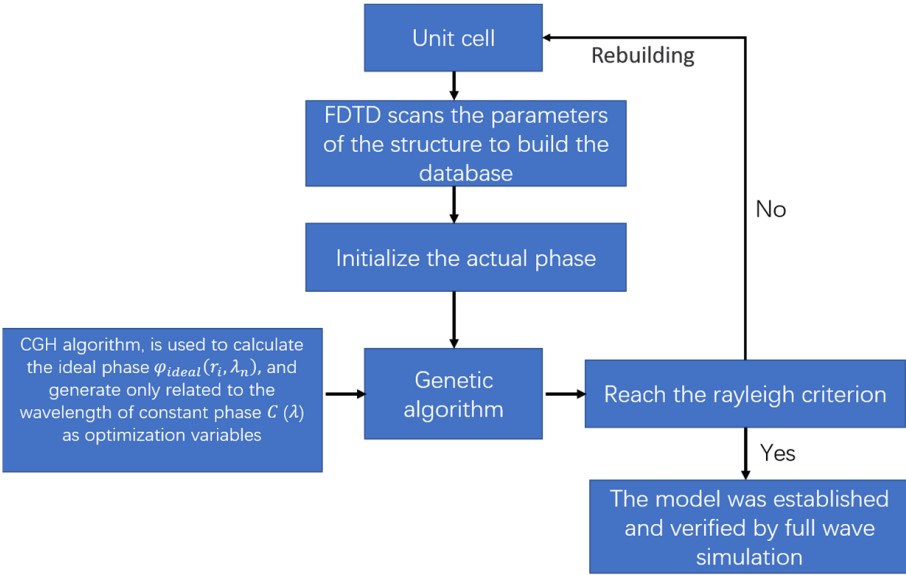

**Figure 3.** Application of genetic algorithm optimization flow chart in design principle.

The independent phase shift of each wavelength provides the possibility to realize wavelength multiplexing devices. However, as can be seen from Figure 2a, this phase shift does not cover every corner of the phase space, and it is necessary to maximize the use of this phase space to achieve the required metasurface. Therefore, a constant phase $C(\lambda)$, which is only related to wavelength, is introduced as the optimization variable. The difference between the ideal phase of different wavelengths and the propagation phase of the structure itself is taken as the objective function, and the genetic algorithm is used to optimize it, so that the phase of the structure itself matches the ideal phase of different

wavelengths to a certain extent. Since it only changes the absolute phase of each wavelength, and not the relative phase, it will not affect the objective function.

$$\Delta\varphi(\lambda_n) = \beta_n \sum_i |\varphi_{ideal}(r_i, \lambda_n) + C(\lambda) - \varphi_{real}(r_i, \lambda_n)| \tag{1}$$

$\varphi_{real}$ represents the propagation phase randomly selected from the phase library, $\varphi_{ideal}$ represents the ideal phase obtained by calculation, $\Delta\varphi$ represents the sum of phase differences of each wavelength, and $\beta_n$ represents the weight factor adjusted by the objective function of different wavelengths based on the simulation results. By changing the weight factor, the relative balance of phase matching of different wavelengths can be achieved.

## 3. Results

### 3.1. Chromatic Dispersion Manipulation

Figure 4a–c show a metalens (32 μm × 32 μm) that arbitrarily controls the dispersion of three wavelengths in the visible region (486 nm, 587 nm, and 656 nm). As a validation of the design scheme, the metalens exhibits achromatic focus for all three wavelengths at two focal lengths of 40 μm and 60 μm. The corresponding phase distribution with dual focal length features designed by us satisfies the following functions:

$$E(x, y, \lambda) = \sum_n e^{-i\left(\frac{2\pi}{\lambda}\left(\sqrt{x^2+y^2+f_n{}^2}-f_n\right)\right)} \tag{2}$$

$$\Phi(x, y, \lambda) = arg(E(x, y, \lambda)) \tag{3}$$

where $f_n$ represents the focal length, $(x, y)$ represents the position of each nanopillar on the metasurface, and $\Phi(x, y, \lambda)$ represents the ideal phase of the holographic metasurface with different wavelengths.

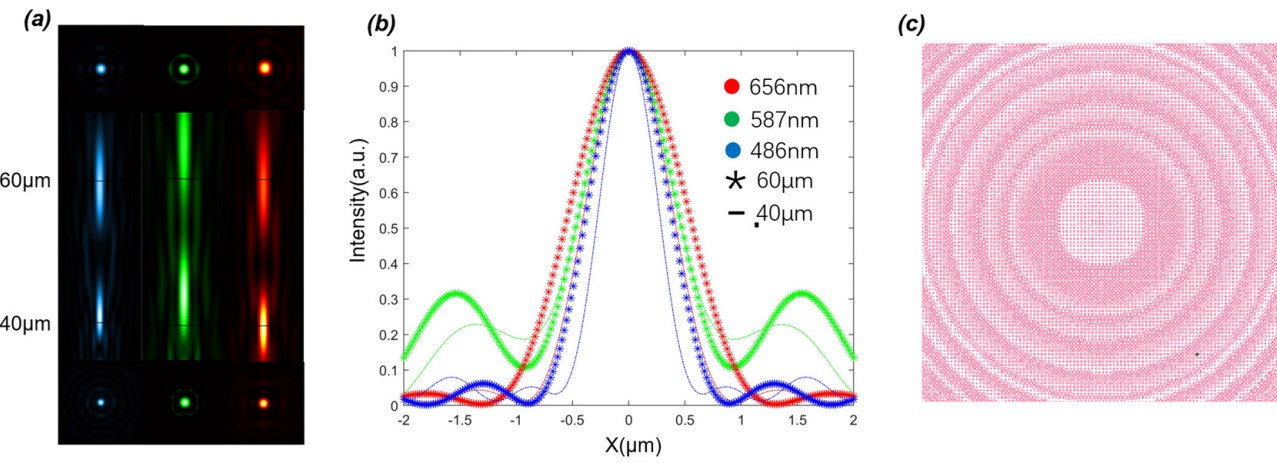

**Figure 4.** (**a**) Longitudinal intensity distribution and transverse intensity distribution simulated by B, G, and R wavelengths; (**b**) Transverse intensity distribution curve corresponding to the corresponding focal position; (**c**) The gds layout of the metalens structure.

The right part of Figure 4a–c represents the gds layout of the metalens structure. The simulation results of corresponding wavelengths are shown on the left. The three wavelengths are blue (486 nm), green (587 nm), and red (656 nm), respectively, which vividly show the longitudinal and transverse intensity distributions.

The corresponding values are shown in Table 1. We can see that the focal plane is consistent with the theoretical expectation, which proves that we have successfully controlled the chromatic aberration.

**Table 1.** Focal position, efficiency, and FWHM of metasurface.

| Wavelength (nm) | Focal Position | Efficiency (%) | FWHM (µm) |
|---|---|---|---|
| 486 nm | 40 µm | 10.34% | 616 nm |
| | 60 µm | 12.98% | 824 nm |
| 587 nm | 40 µm | 26.59% | 968 nm |
| | 60 µm | 29.96% | 968 nm |
| 656 nm | 40 µm | 29.66% | 856 nm |
| | 60 µm | 29.15% | 1144 nm |

*3.2. Focused Beam Array*

In order to generate the focal spots of the array in the focal plane, we use the hologram superposition principle, which can integrate the information of each focal spot in two-dimensional space. The superimposed complex field distribution can be expressed as Equation (4):

$$E(x, y, \lambda) = \sum_m \sum_n A_{mn} e^{-i\left(\frac{2\pi}{\lambda}\left(\sqrt{(x-x_m)^2+(y-y_n)^2+f^2}-f\right)\right)} \tag{4}$$

where $f$ represents the focal length, $(x, y)$ represents the position of each nanopillar on the metasurface, $(x_m, y_n)$ represents the position of each focal spot, and $A_{mn}$ represents the amplitude distribution of the corresponding focal spot. In order to simplify the calculation, we set R, G, and B as the wavelengths of the incident light wave, and here we select 656 nm, 587 nm, and 486 nm to represent R, G, and B, respectively. For multi-channel beam phase information, use the CGH algorithm. As shown in Equation (5), $\Phi(x, y, \lambda)$ represents the ideal phase of the holographic metasurface with different wavelengths:

$$\Phi(x, y, \lambda) = arg(E(x, y, \lambda)) \tag{5}$$

As can be seen from Figure 5, after optimization, the matching between the real phase and the ideal phase is achieved to a certain extent. Through the calculation of Equation (6), we can determine that $\Delta P(\lambda_1) = 0.5681$ rad, $\Delta P(\lambda_2) = 0.7812$ rad, and $\Delta P(\lambda_3) = 0.7643$ rad. It is noted that the increased phase errors versus the related wavelengths are due to the insufficient phase coverage at long wavelengths. However, according to the Rayleigh criterion, when the wave aberration between the actual wave surface and the ideal wave surface is less than $\lambda/4$, the actual wave surface can be regarded as defect-free [37]. Moreover, it is noted that if the defect part takes a small proportion in the whole wave surface area, these local defects can still be ignored even if the wave aberration is greater than $\lambda/4$. Our phase error value is much smaller than the criterion, which proves the rationality of our design from the perspective of phase. Next, we will perform full-wave simulation verification through simulation software.

$$\Delta P(\lambda_n) = \left\{ \sum_i |\varphi_{ideal}(r_i, \lambda_n) + C(\lambda) - \varphi_{real}(r_i, \lambda_n)| \right\} / n_0 \tag{6}$$

where $\Delta P(\lambda_n)$ represents the average phase error between the ideal phase and the propagation phase of each wavelength and $n_0$ represents the number of unit structures contained in the metasurface.

We designed a metasurface with a diameter of 32 µm × 32 µm and a focal length of 10 µm. The focusing field results of B, G, and R wavelengths are obtained by full-wave simulation software. It can be seen from Figure 6 that the position arrangement of the focal spot is almost the same as the theoretical design (4 microns from the origin, hexagonal distribution), the maximum position error is only about 0.31 λ, and the stray light is small and the energy is concentrated.

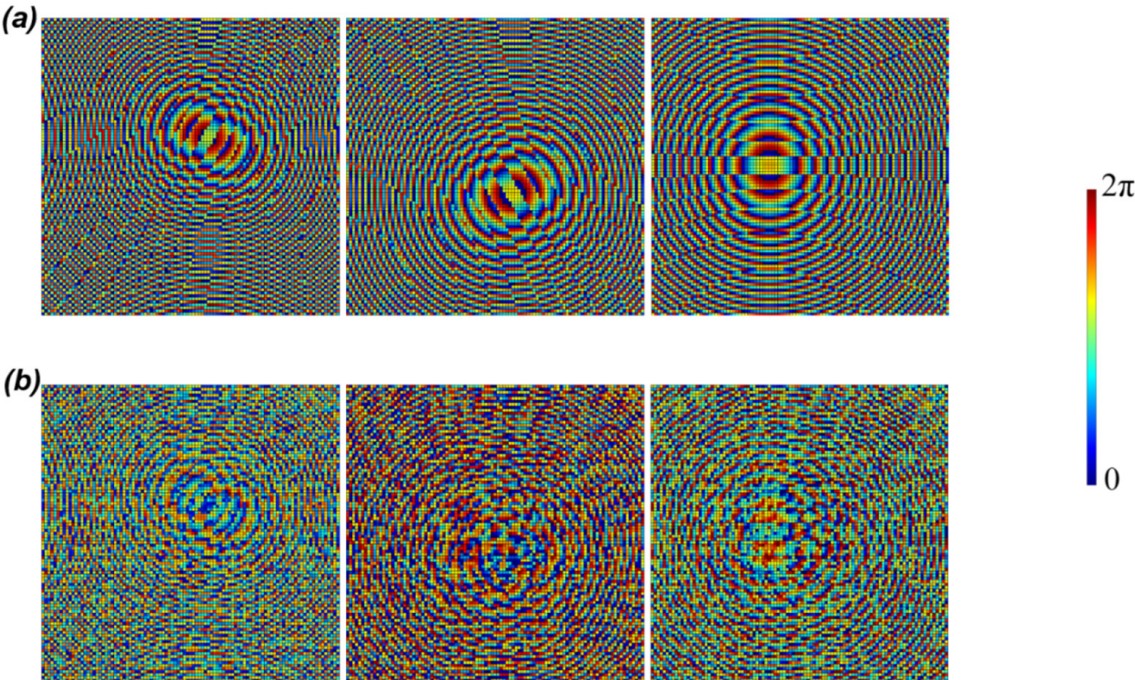

**Figure 5.** (**a**) From left to right, the ideal phase distributions corresponding to the three wavelengths B, G, and R, respectively, are shown; (**b**) From left to right, the real phase distributions corresponding to the three wavelengths B, G, and R, respectively, are shown.

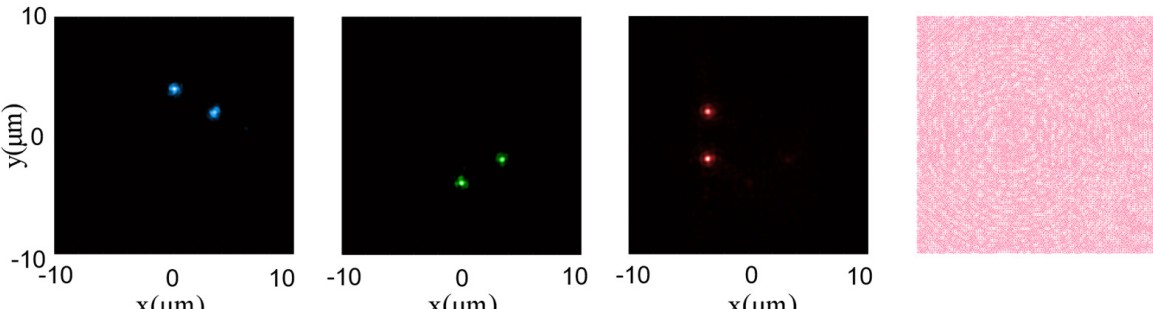

**Figure 6.** The first three images from left to right, respectively, represent the normalized intensity distribution of the focal planes of B, G, and R waves, while the final image is the gds layout of the metalens structure.

### 3.3. Vortex Beam Array

In the fields of communication, control of microscopic particles, and superresolution imaging, vortex beams are of undoubted importance. The array form of the vortex beam is more helpful to improve its efficiency and freedom. In this section, we still apply the design method in Section 3 to design a multi-channel vortex beam generator with wavelength multiplexing, and the wavelength is the same as the selection in Section 3.1. The hologram superposition principle is still adopted and the complex amplitude distribution of the vortex array can be expressed as Equation (7):

$$E(x,y,\lambda) = \sum_m \sum_n A_{mn} e^{-i\left(\frac{2\pi}{\lambda}\left(\sqrt{(x-x_m)^2+(y-y_n)^2+f^2}-f\right)\right)} * e^{i(l_{mn}\phi_{mn}(x-x_m,y-y_n))} \qquad (7)$$

The meaning of the parameters in Equation (7) is virtually the same as that in Equation (4). In particular, $l_{mn}$ represents the topological charge number of each vortex in the array and $\phi_{mn}(x-x_m, y-y_n)$ represents the azimuth distribution of the unit cell after relative displacement. The phase information of the multi-channel vortex beam is still obtained by CGH, as

shown in Equation (5), which represents the ideal phase of the holographic metasurface with different wavelengths.

As can be seen from Figure 7, after optimization, the matching between the real phase and the ideal phase of the vortex array is also achieved to a certain extent. The calculated values of $\Delta P(\lambda_1) = 0.5859$ rad, $\Delta P(\lambda_2) = 0.766$ rad, and $\Delta P(\lambda_3) = 0.7997$ rad prove that the error between our ideal wave surface and the actual wave surface is small. The simulation software was again used for full-wave simulation verification.

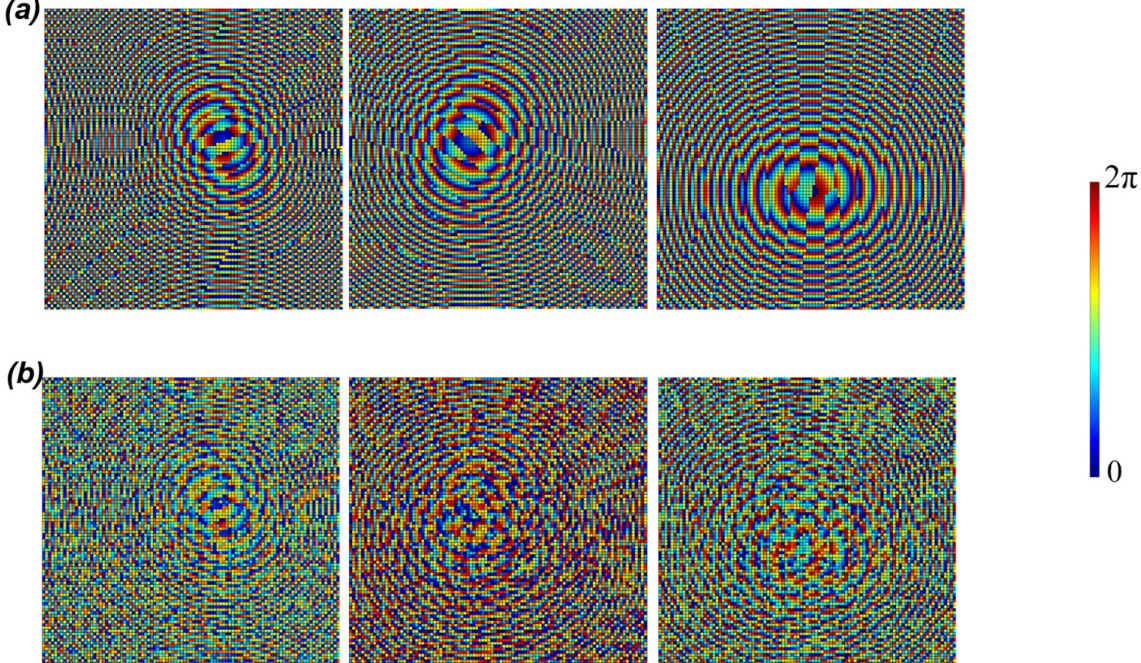

**Figure 7.** (**a**) From left to right, the ideal phase distributions of the vortex array corresponding to wavelengths B, G, and R, respectively, are shown; (**b**) From left to right, the real phase distributions of the vortex array corresponding to wavelengths B, G, and R, respectively, are shown.

It can be seen from Figures 8 and 9 that under the illumination of light waves of different wavelengths, different vortex arrays are emitted from the metasurface, and the topological charge of the vortex array is 1. The simulation results are consistent with the theoretical design. However, this design cannot always produce good results. When the number of light spots generated is too large, especially when there are different patterns of light spots mixed together, the energy distribution of the light spots may be uneven, and there may be other diffraction spots.

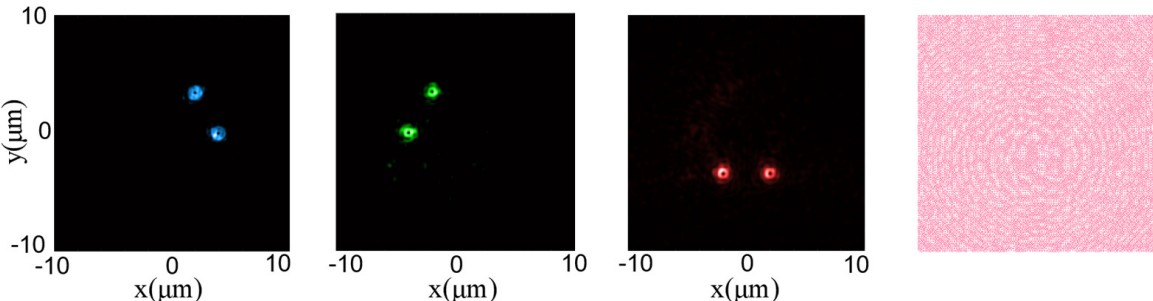

**Figure 8.** The first three images from left to right, respectively, represent the normalized intensity distribution of the focal planes of B, G, and R waves. The final image is the gds layout of the metalens structure.

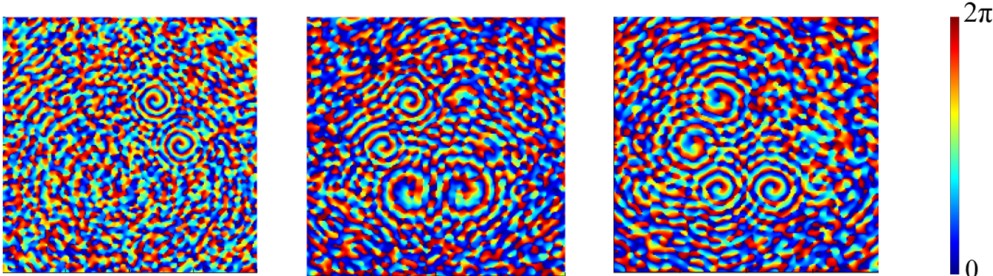

**Figure 9.** Phase profiles of the beam generated by B, G, and R wavelengths.

From Figure 10, we can see that with the changes in the number and mode of light spots, the uneven energy distribution of light spots becomes increasingly obvious, and there are other diffraction spots. To solve this problem, we can balance the energy intensity between each light spot by changing Equation (7) and regulate the phase matching degree of each wavelength by changing the value in Equation (1), in order to weaken the intensity of the interference diffraction spot.

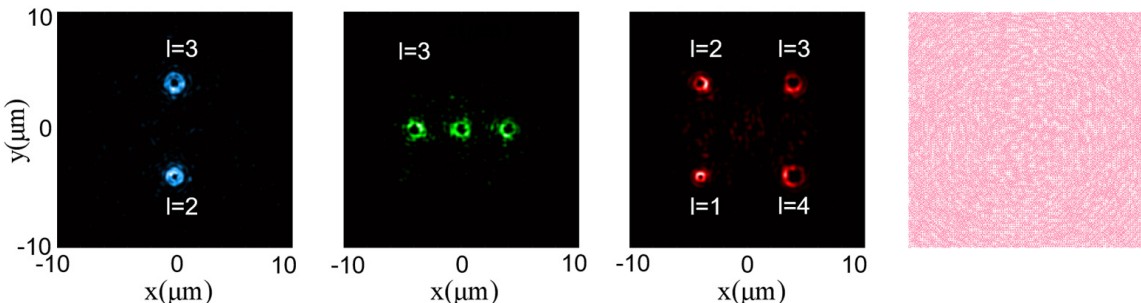

**Figure 10.** The first three images from left to right, respectively, represent the normalized intensity distribution of the focal planes of B, G, and R waves. The final image is the gds layout of the metal-ens structure.

## 4. Discussion

In conclusion, for a given trichromatic wavelength, the holographic metasurface structure with three different wavelengths multiplexing is implemented in this paper, and the beam array with a different arrangement is obtained. The simulation results show that by adjusting the weight factor $\beta_n$ in Equation (1) and then changing the phase error value $\Delta P(\lambda_n)$ in Equation (6), the interference of diffraction spots can be significantly weakened. Of course, this requires that the wave aberration between the ideal wave surface and the actual wave surface meets the Rayleigh criterion. By changing $A_{mn}$ in Equations (4) and (7), the light field intensity of each spot in the array can be regulated. However, this regulation is limited by many factors. When the metasurface area is fixed, factors such as the proportion of light field intensity at each spot need to be considered comprehensively. The intensity of each spot will be correspondingly reduced. Meanwhile, there may be adverse factors such as interference diffraction spots around the array. In particular, when the selected wavelength difference is small, it will lead to obvious secondary focus interference. From the simulation results, when the selected wavelength difference is greater than 100 nm, the relevant interference can be better avoided and the desired beam quality can be obtained.

The wavelength multiplexing holographic metasurface structure design method is expected to be extended to the design of 3D metasurface beam arrays and will obtain better beam controllability, which provides a new way for the acquisition of polarization-free multi-information from array beams such as imaging and light capture and improves information capacity and security.

**Author Contributions:** Conceptualization, T.Z.; model, T.Z.; numerical simulation, T.Z. and Y.X.; investigation, Y.W. (Yue Wang) and J.W.; formal analysis, Y.W. (Yihui Wu) and J.W.; writing—original draft preparation, T.Z. and Y.W. (Yihui Wu); writing— review and editing, T.Z., W.Z. and Y.W. (Yihui Wu); project administration, Y.W. (Yihui Wu); resources, Y.W. (Yihui Wu). All authors have read and agreed to the published version of the manuscript.

**Funding:** National Natural Science Foundation of China (NSFC) (U21A20395, 61727813).

**Institutional Review Board Statement:** Not applicable.

**Informed Consent Statement:** Not applicable.

**Data Availability Statement:** The data supporting the findings of this study are available within the article.

**Conflicts of Interest:** The authors declare no conflict of interest.

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
