# Peer review of "Design of the Polarization-Independent Wavelength Multiplexing Holographic Metasurface"

_photonics, doi:10.3390/photonics10020139_

Round 1

Author Response

Dear Reviewer,

Thank you for taking the time to review our manuscript entitled “Design of the Polarization Independent Wavelength Multiplexing Holographic metasurface”. We have carefully considered your comments and have made the following changes to the manuscript:

Your major concern:

In response to your first and second comments:

Our answer is given in lines 208 to 230 of the Revised Paper. As can be seen from Figure 8-9, when the number of oam modes generated is consistent and small, relatively good results can be obtained. In order to solve the problem of uneven energy distribution of light spots and interference with diffraction spots, we can adjust Amn in Equation 7 and βn in Equation 1 respectively to weaken them. Generally, if the number of modes is high, the energy is relatively weak, and a higher  Amn can be allocated. For the wavelength with serious interference diffraction spots, a higher βn can be assigned to improve the degree of matching between the ideal phase and the actual phase.

Your question is very good, so I have spent quite a long time to analyze it. To some extent, the pure phase regulation method I chose is not the optimal method, because it does not consider the factor of amplitude[1]. The problem in reference 1 is similar to ours, except that he uses geometric phase regulation method. The greatest advantage of geometric phase is that its amplitude and phase are separate and can be adjusted separately. However, the unit structure we selected is cylindrical and titanium dioxide is used as material, which results in the transmittance of our unit structure is almost above 90% in the visible band. This is an advantage to some extent, but a disadvantage for those who want to use a mixture of amplitude and phase modulation. Originally, I wanted to change the material to silicon for a trial, but it took too long to build the database and the time to reply to comments was limited, which eventually led to the aborted idea. If you are very serious about this problem, I can continue to try. Thank you again for your question.

Minor but still important comments:

1-6 questions to solve according to your requirements. There are many mistakes that should not be made. Thank you very much for pointing out and correcting them.

  1. Your opinion is quite reasonable, but I'd like to stick to my own opinion. Since I was working on a series of different metasurface structures, I needed to use the gds map to show the difference and authenticity of the structures. Of course, if you are serious about this problem, I will modify it.
  2. From all the literature I have read so far, no metasurface with multiwavelength multiplexing has yet achieved a focusing efficiency of >90%, including polarization related ones. As far as my method is concerned, there are many ways to improve efficiency. Such as 1. Increase the size of the lens (more structures will make the light more concentrated), 2. Increase the height of the cell structure (the higher the cell structure, the stronger the structure's ability to bind light, means the more similar the structure is to a truncated waveguide, the weaker the coupling between the structures), 3. Increase the refraction index of the structure itself (for reasons similar to 2)

In fact, when most people adopt this similar method, they try to avoid the visible band, the reason is that it is impossible to choose a more appropriate unit structure, so most of them choose to work in the infrared band, and the material chooses silicon with higher refractive index and higher height as the material.

Because my design product needs to work in the visible band, I choose titanium dioxide, which is commonly used in the visible band, as the material. At the same time, considering the precision of titanium dioxide processing in China, we choose 600nm as its height. It may be a good idea to continue to increase the size of the lens, but larger size means better computer configuration and longer time consumption. Generally, people can not carry out full-wave simulation verification after choosing a larger size, and can only calculate the results approximated by the vector angular spectral diffraction theory. So with that in mind, I went with today's structure.

9-10 Error corrected

  1. The corresponding front view has been changed to the structure's gds layout
  2. The corresponding roles of and have been added in the article, and the conclusion has also been adjusted
  3. Error corrected
  4. Yes, there are wavelength multiplexed polarization-dependent metasurfaces in the ultraviolet field, using aluminum nitride[2]. The corresponding result can be achieved only by converting the corresponding geometric phase into the corresponding transmission phase. The same goes for the X-ray band
  5. The correct grammar was revised in the “formal paper”. The "Revise paper" acted as a reference.

We hope that these changes address your concerns and that you will find the revised manuscript suitable for publication.

We appreciate your time and effort in reviewing our manuscript.

Sincerely,

Tianyu zhao

references

1 Jin, J., Pu, M., Wang, Y., Li, X., Ma, X., Luo, J., ... & Luo, X. (2017). Multi‐channel vortex beam generation by simultaneous amplitude and phase modulation with two‐dimensional metamaterial. Advanced Materials Technologies2(2), 1600201.

2 Guo, L., Hu, Z., Wan, R., Long, L., Li, T., Yan, J., ... & Wang, L. (2019). Design of aluminum nitride metalens for broadband ultraviolet incidence routing. Nanophotonics8(1), 171-180.

Reviewer 2 Report

In this manuscript, the authors proposed a design for wavelength multiplexing holographic metasurface with polarization independence. Overall, the manuscript is well organized. I would recommend it be published on Photonics if the following comments can be addressed and the English writing can be improved.

1. The authors should cite references to support several claims in the paper. (Page2, lines 59-64)

2. Left figure in Figure 1 is confusing: it looks like different focal points from different wavelengths were generated from different segmentation of the metasurface - this requires spacially distributed color filters, which is not the case demonstrated in this paper lately. 

3. Is the high aspect ratio TiO2 nano pillar (D80nm/H600nm) fabricatable?

4. What does "this phase shift does not cover every corner of the phase space" mean on Page 3, lines 98-100?

5. Figure4(a) should have 40um and 60um in the y-axis instead of nm in units.

6. There is an obvious chromatic aberration in the green light shown in Figure 4 (a) - the focal points are not the same as blue and red light. This result is not supporting the authors' claim on Page5 lines 146-148. Please explain. 

7. The results shown in Figure 8 do not indicate the topological charges of the generated vortex - the donut-shaped intensity distribution is not enough to characterize OAM, please add the phase profile of the generated beam to support the authors' claim.

Author Response

Dear Reviewer,

Thank you for taking the time to review our manuscript entitled “Design of the Polarization Independent Wavelength Multiplexing Holographic metasurface”. We have carefully considered your comments and have made the following changes to the manuscript:

  1. The references section has been updated. Thanks for the reminder.
  2. The result of our research is to construct different beam arrays, but the current research on metasurface color routers can only realize the focused beam of different positions generated by different wavelengths. Our results are a further extension of this field in terms of the function of phase composition and realization. To date, no similar results have been found.
  3. It is manufacturable. We have consulted Shanhe Optoelectronic Co., LTD in Suzhou, China, and their answer is yes.
  4. There are different ideal phases for different wavelengths, and we have only one structure, which means we need to find a structure whose transmission phases at different wavelengths need to correspond to the ideal phase of each wavelength. However, it is not always possible to find such a structure that the transmission phase of the three wavelengths matches the ideal phase of the corresponding wavelength, that is, the constituted transmission phase space is insufficient.
  5. Error corrected
  6. Based on wave optics theory, the focused light field has a certain focal depth along the axial direction(DOF):

         DOF=λ/NA ( λ is the wavelength and NA is the numerical aperture)  (1)

         DOF>=abs(f(ω1)-f(ω2))                                                                     (2)

If the absolute value of the focal length difference of two frequencies (ω1,ω2) is less than the focal depth corresponding to one of them, it can be considered that even if the metalens designed for a single frequency , there is basically no chromatic difference for the other frequency [1]. Our design is in line with this standard.

When the number of wavelengths involved is too large or the wavelength difference is too large, our design results will indeed deteriorate. At this time, we can choose a variety of shapes of the unit structure to expand our database (of course, the premise is that the refractive index of the structural unit is very large, the structure is very high, so as to reduce the coupling between the unit structure), so as to weaken the influence of chromatic aberration and other factors.

Due to time constraints and manufacturing problems with the titanium dioxide itself, we did not choose the above method.

  1. Our answer is given in lines 208 to 230 of the Revised Paper. As can be seen from Figure 8-9, when the number of oam modes generated is consistent and small, relatively good results can be obtained. In order to solve the problem of uneven energy distribution of light spots and interference with diffraction spots, we can adjust Amn in Equation 7 and βn in Equation 1 respectively to weaken them. Generally, if the number of modes is high, the energy is relatively weak, and a higher Amn can be allocated. For the wavelength with serious interference diffraction spots, a higher βn can be assigned to improve the degree of matching between the ideal phase and the actual phase.

Your question is very good, so I have spent quite a long time to analyze it. To some extent, the pure phase regulation method I chose is not the optimal method, because it does not consider the factor of amplitude[2]. The problem in reference 2 is similar to ours, except that he uses geometric phase regulation method. The greatest advantage of geometric phase is that its amplitude and phase are separate and can be adjusted separately. However, the unit structure we selected is cylindrical and titanium dioxide is used as material, which results in the transmittance of our unit structure is almost above 90% in the visible band. This is an advantage to some extent, but a disadvantage for those who want to use a mixture of amplitude and phase modulation. Originally, I wanted to change the material to silicon for a trial, but it took too long to build the database and the time to reply to comments was limited, which eventually led to the aborted idea. If you are very serious about this problem, I can continue to try. Thank you again for your question.

  1. The corresponding grammatical errors have been corrected in the paper

We hope that these changes address your concerns and that you will find the revised manuscript suitable for publication.

We appreciate your time and effort in reviewing our manuscript.

Sincerely,

Tianyu zhao

references

1 肖行健, 祝世宁(Academician of Chinese Academy of Sciences), & 李涛. (2021). 离散波长消色差超构透镜的性能分析. 中国光学14(4), 823.

2 Jin, J., Pu, M., Wang, Y., Li, X., Ma, X., Luo, J., ... & Luo, X. (2017). Multi‐channel vortex beam generation by simultaneous amplitude and phase modulation with two‐dimensional metamaterial. Advanced Materials Technologies2(2), 1600201.

Round 2

Reviewer 2 Report

The authors' reply addressed my concern well. I recommend this manuscript to be published at Photonics.